# Developing an Interpretable Machine Learning Model to Predict in-Hospital Mortality in Sepsis Patients: A Retrospective Temporal Validation Study

**DOI:** 10.3390/jcm12030915

**Published:** 2023-01-24

**Authors:** Shuhe Li, Ruoxu Dou, Xiaodong Song, Ka Yin Lui, Jinghong Xu, Zilu Guo, Xiaoguang Hu, Xiangdong Guan, Changjie Cai

**Affiliations:** 1Department of Critical Care, The First Affiliated Hospital of Sun Yat-Sen University, Guangzhou 510080, China; 2Department of Anesthesiology, The First Affiliated Hospital of Sun Yat-Sen University, Guangzhou 510080, China; 3Department of Statistics, William March Rice University, 6100 Main St, Houston, TX 77005, USA

**Keywords:** sepsis, mortality, extreme gradient boosting, temporal validation, interpretability

## Abstract

Background: Risk stratification plays an essential role in the decision making for sepsis management, as existing approaches can hardly satisfy the need to assess this heterogeneous population. We aimed to develop and validate a machine learning model to predict in-hospital mortality in critically ill patients with sepsis. Methods: Adult patients fulfilling the definition of Sepsis-3 were included at a large tertiary medical center. Relevant clinical features were extracted within the first 24 h in ICU, re-classified into different genres, and utilized for model development under three strategies: “Basic + Lab”, “Basic + Intervention”, and “Whole” feature sets. Extreme gradient boosting (XGBoost) was compared with logistic regression (LR) and established severity scores. Temporal validation was conducted using admissions from 2017 to 2019. Results: The final cohort included 24,272 patients, of which 4013 patients formed the test cohort for temporal validation. The trained and fine-tuned XGBoost model with the whole feature set showed the best discriminatory ability in the test cohort with AUROC as 0.85, significantly higher than the XGBoost “Basic + Lab” model (0.83), the LR “Whole” model (0.82), SOFA (0.63), SAPS-II (0.73), and LODS score (0.74). The performance in varying subgroups remained robust, and predictors, such as increased urine output and supplemental oxygen therapy, were crucially correlated with improved survival when interpretability was explored. Conclusions: We developed and validated a novel XGBoost-based model and demonstrated significantly improved performance to LR and other scores in predicting the mortality risks of sepsis patients in the hospital using features in the first 24 h.

## 1. Introduction

Sepsis is life-threatening organ dysfunction that occurs due to a dysregulated host response to infection [1]. Sepsis represents a leading cause of death in intensive care units (ICU), as there is an annual rate of 48.9 million cases of sepsis worldwide, of which 11.1 million patients died, comprising 19.7% of all deaths globally [2]. Early prognostic prediction in sepsis patients plays a vital role in bedside decision making [3]. For instance, patients who are at low mortality risk tend to receive standard care, whereas high-risk patients might gain more benefits from aggressive therapies carrying more than minimal risk. Prognostic prediction can serve as a tool in study design for preferentially selecting high-risk patients and reducing the size of enrolment [4]. In addition, risk prediction models may be used in patient counseling to initiate the discussion about palliative care, and also may be used for quality-of-care outcome assessment and benchmarking [5].

Single serum biomarkers, such as procalcitonin, interleukin-6 [6], presepsin [7], or CD64 [8], have shown insufficient sensitivity or specificity in practice, because sepsis is a heterogeneous clinical syndrome presenting with varying infection etiologies, unique individual genetics, and distinctive host–pathogen interactions [4,9]. On the other hand, traditional scoring systems face new challenges nowadays. For instance, Sequential Organ Failure Assessment (SOFA) score was initially created based on expert consensus [1]. Acute Physiology and Chronic Health Evaluation (APACHE) score used binary logistic regression (LR) equations [10], requiring no intercorrelation between features and displaying a limited ability to deal with missing data [11]. However, multicollinearity and missingness are frequently observed in real-world data.

In the era of electronic health data explosion and machine learning (ML) algorithm advancement, employing data-driven ML-based systems in daily ICU practice has signaled a promising future in healthcare [12]. Researchers have already developed ML-based predictive approaches in critically ill patients with sepsis [13,14], influenza [15], cerebral hemorrhage [16], etc. Nevertheless, the clinical impact of these models has been heavily limited, and one of the main reasons is that only a small number of them are validated externally to ensure reproducibility and generalizability [17]. Another significant obstacle is the “black-box” nature of such systems, especially in critical fields, such as healthcare [18]. Good interpretability has become a crucial issue in model development, not only for researchers to verify the reliability of new models, but also for clinicians to gain confidence in sharing decision-making responsibilities with understandable algorithms [12]. Novel knowledge generated from interpretable models can also bring new insights into routinely collected clinical data.

Therefore, we aimed to develop an interpretable ML model to predict in-hospital mortality in sepsis patients, utilizing highly dimensional real-world data from the Medical Information Mart for Intensive Care (MIMIC)-IV database. A robust supervised algorithm, extreme gradient boosting (XGBoost), was implemented and compared with the LR model and established severity scores. Moreover, we used an independent test dataset for temporal validation, and employed visualized interpretation of our model.

## 2. Methods

### 2.1. Database and Ethical Approval

The study utilized an up-to-date open-source critical care database MIMIC-IV (Version 2.1) [19,20]. The database enrolled over fifty thousand unique patients admitted at the Beth Israel Deaconess Medical Center (BIDMC) from 2008 to 2019 in the United States (inclusive). This newly released version contained highly granular clinical data, including hourly recorded vital signs, laboratory measurements, intake and output volumes, calorie and protein intakes of enteral and intravenous medications, etc. 

The Institutional Review Board of the BIDMC and Massachusetts Institute of Technology approved the research use of the MIMIC database [21]. One author (S. Li) completed the Collaborative Institutional Training Initiative training program “Human Research, Data or Specimens Only Research” to gain access to the database (Record ID: 32396061). Therefore, the requirement for individual patient consent was waived because of the deidentified nature of MIMIC-IV. All methods were executed under relevant guidelines and regulations.

### 2.2. Cohort and Feature Selection

Cohort selection was executed according to the third international consensus definition, Sepsis-3 [1]. Sepsis was screened, satisfying suspected host infection (determined by ordering a microbiology culture or antibiotics), accompanied by SOFA score >= 2 during the hospitalization. Patients under 18 years old, or having repetitive ICU admissions except for the first time, or staying in ICU for less than 24 h were excluded.

We performed data extraction with PostgreSQL (Version 14.0; PostgreSQL Global Development Group). The profile of 125 features is summarized in Appendix A, including demographic characteristics, vital signs, laboratory measurements, and clinical interventions within 24 h after ICU admission. 

Additionally, Simplified Acute Physiology Score (SAPS)-II [22], APS-III [23], Logistic Organ Dysfunction Score (LODS) [24], Oxford Acute Severity of Illness Score (OASIS) [25], and SOFA score [26] acted as benchmarks for evaluating the performance of the new scoring tool. A detailed description of variable and cut-off points selected in the severity scores utilized in this study is displayed in Appendix A.

The outcome was all-cause mortality occurring in the hospital after the first day in the ICU; thus, the prediction window was 24 h.

### 2.3. Statistical Analysis and Missing Data Processing

All statistical analysis and machine learning tasks were conducted on R (Version 4.1.3; R Foundation for Statistical Computing), and wielded packages included dplyr [27], ggplot2 [28], arsenal [29], mice [30], VIM [31], car [32], caret [33], Matrix [34], xgboost [35], SHAPforxgboost [36], and pROC [37]. Univariate analysis was performed between survivors and non-survivors. Continuous features were described as mean (SD) or median (Q1, Q3) based on the normality of distribution (checked by normal probability plots), and compared by the Student’s *t*-test or Wilcoxon rank-sum test as appropriate. Categorical features were presented as frequency (percentage) and compared by Pearson’s chi-square test. A two-sided *p* < 0.05 was considered statistically significant. 

Missing values mainly consisted of infrequently ordered laboratory measurements, which can be classified as missing at random (MAR) and imputed. Features with absent proportions above 40% were excluded from modeling [38]. Nevertheless, liver function tests, such as alanine transaminase and total bilirubin, were preserved despite the missing proportions of 41–42%, because dropping these features would cause a huge neglection in liver function assessment. Missing records were imputed based on predictive mean matching [39]. The absence of maximal vasopressor rates was regarded as non-usage and imputed with zero. 

### 2.4. Machine Learning Model and Evaluation

Based on the longitudinal information provided from MIMIC-IV, the development cohort only included patients admitted from 2008 to 2016, split into training and validation dataset by stratified random sampling according to hospital mortality (ratio 8:2). In addition, we performed a temporal validation using patients admitted from 2017 to 2019 as a holdout test dataset.

Next, we employed XGBoost and multivariate LR models to predict the probability of in-hospital mortality. XGBoost is a decision-tree-based algorithm, where weak learners are ensembled iteratively into a strong predictor when the loss function is minimized through gradient boosting [40]. All features were re-classified into different genres and utilized to construct models under the following combinations. “Basic + Lab” strategy included demographic characteristics, vital signs, and laboratory measurements; “Basic + Intervention” strategy included treatments administered on the first day rather than laboratory tests; “Whole” strategy used all mentioned features. Features included in the final models are summarized in Appendix A. To optimize the XGBoost model, we then executed hyperparameter tuning through five-fold cross-validation (CV). Optimal values were obtained through 20 rounds of CV iterations in the previously defined ranges, and the final parameters were fed into the new model (Appendix A).

Subsequently, the discriminatory ability was assessed using the area under the receiver operating characteristic curve (AUROC), sensitivity (Se), specificity (Sp), accuracy, positive predictive value (PPV), negative predictive value (NPV), and diagnostic odds ratio (DOR) at cut-off points. DeLong method was used to calculate 95% confidence interval (CI) of AUROC and to compare with LR and single severity scores. Calibration curve was plotted to assess the agreement between estimated and actual probabilities. Feature importance and Shapley Additive explanation (SHAP) summary plots were illustrated to display feature contributions and potential impacts on in-hospital mortality risks using the “Whole” model. In addition, model performance was evaluated on different subgroups, including admission type, admitted ICU type, and source of infection.

## 3. Results

### 3.1. Patient Characteristics and Missingness

In total, 24,272 patients fulfilling the criteria were included (Figure 1), of whom 15.5% died in hospital (N = 3759). Demographic and clinical features between survivors and non-survivors are compared in Table 1 and Appendix A. Non-survivors tended to be older (*p* < 0.001), acquire higher severity scores (*p* < 0.001), and more frequently admitted through the emergency department (*p* < 0.001). They also presented significantly worse vital signs and laboratory values, for instance, elevated maximum red blood cell distribution width (RDW) (16.0 vs. 14.7%, *p* < 0.001) and higher minimum blood urea nitrogen (BUN) (27.0 vs. 18.0 mg/dL, *p* < 0.001). Non-survivors were more likely to receive invasive ventilation (48.2% vs. 39.8%, *p* < 0.001) and renal replacement therapy (6.3% vs. 2.6%, *p* < 0.001), when the prevalence of supplemental oxygen therapy remained significantly lower (35.4% vs. 53.3%, *p* < 0.001). The hospital-staying days in those who died in the hospital were significantly shorted than surviving patients (6.5 vs. 8.1 days, *p* < 0.001) (Figure 2A). Moreover, 91.5% (22,202) of all patients had sepsis attacks in the first 24 h, and non-survivors experienced a more acute sepsis onset than survivors following ICU admission (2.0 vs. 3.2 h, *p* < 0.001) (Figure 2B). 

Among all 125 features, 26 features had missing proportions above 40%, of which 8 features, relating to liver function, and 6 features of maximal vasopressor rate were preserved, and 12 features were subsequently discarded (Appendix A). The missing patterns were not uniform based on individual patients (Appendix A). Ten iterations of multiple imputation were executed in incomplete features. The imputation process of liver function measurements was visualized due to the relatively high missing proportions (Appendix A), indicating a matched distribution of imputed values with existing ones. 

### 3.2. Model Performance

In the training cohort (N = 16,208), LR and XGBoost both showed good discrimination, when the AUROC of the XGBoost “Basic + Lab” model was the highest (0.97, 95% CI: 0.964–0.971). Among the three types of feature combinations, the “Basic + Intervention” models demonstrated significantly lower performance compared to the other two feature sets (*p* < 0.001). Whole-feature selection strategy resulted in the best discrimination among three LR models (AUROC: 0.84, *p* < 0.001), whereas the performance of this combination was significantly lower than “Basic + Lab” strategy when XGBoost was used (*p* < 0.001).

Moreover, in the independent test cohort consisting of patients admitted at a later period for temporal validation (N = 4013), XGBoost models all exhibited appealing discriminatory ability compared to LR models, utilizing feature combinations “Basic + Lab” (AUROC: 0.83 vs. 0.80, *p* < 0.001), “Basic + Intervention” (AUROC: 0.80 vs. 0.77, *p* < 0.001), and “Whole” (AUROC: 0.85 vs. 0.82, *p* < 0.001) (Table 2). Furthermore, the XGBoost “Whole” model outperformed well-acknowledged scores for predicting hospital outcomes of sepsis patients, including SOFA score (AUROC: 0.63, *p* < 0.001), SAPS-II (AUROC: 0.73, *p* < 0.001), APS-III (AUROC: 0.74, *p* < 0.001), LODS (AUROC: 0.74, *p* < 0.001), and OASIS (AUROC: 0.70, *p* < 0.001) (Figure 3). DOR of the XGBoost “Whole” model was the highest (10.87), followed by XGBoost “Basic + Lab” model (9.23), suggesting highly efficient predictive tools.

### 3.3. Interpretability and Calibration

The XGBoost model was then visually interpreted to provide compelling insights. Feature importance rankings in the XGBoost “Whole” model implied that predicting in-hospital mortality was heavily dependent on clinical features, such as urine output, maximum norepinephrine rate, maximum RDW, age, etc. (gain: 0.08, 0.04, 0.03, 0.02, respectively) (Appendix A). Moreover, the SHAP summary plot (Figure 4) revealed that poor hospital survival was intensely associated with being elderly, an underlying cardiovascular disorder or cancer, increased levels of RDW and BUN, and impaired pulse oxygen saturation, etc. Nevertheless, increased urine volume and administering supplemental oxygen therapy indicated beneficial signs for sepsis, reflecting relatively normal kidney and respiratory functions. Odds ratios (ORs) of multivariate LR models are displayed in Appendix A. In addition, the calibration plot indicated that the XGBoost “Whole” model had an overall consistent agreement between predictive and observational in-hospital mortality risks (Figure 5), especially when observed mortality risks were below 20%. The model tended to underestimate the mortality risk when the observed risks rose beyond approximately 40%.

### 3.4. Performance in Subgroups

We inspected the novel XGBoost “Whole” model in subgroups with different admission routes, admitted ICU types, and positive microbiological cultures from different sources in the body (Table 3). The model reflected stability in performance across various subgroups with varying mortality rates, from 1.8% to 29.3%. The AUROC in elective patients (N = 163) remained the highest (0.98, 95% CI: 0.930–1.000), when the AUROCs went significantly lower among individuals with definite evidence of respiratory tract (0.72, 95% CI: 0.668–0.771) and gastrointestinal/peritoneal infection (0.72, 95% CI: 0.605–0.843).

## 4. Discussion

In this study, we developed an interpretable machine learning approach for in-hospital mortality prediction in the ICU using baseline characteristics, vital signs, lab measurements, and clinical management details from 24,272 sepsis patients. The XGBoost model with whole feature selection showed significantly better discrimination to LR models and other existing scores, e.g., SAPS-II, SOFA, and LODS. The predictive performance remained robust in temporal validation.

Most acknowledged severity scores are created based on multivariate logistic regression equations [10,24,41]. Traditional LR models are well interpretable and easy to understand. However, they bear limitations, such as a necessity for satisfying linearity relationships, rejecting multicollinearity [11], and the incapacity of handling missing data. Considering the complexity and heterogeneity of sepsis, more sophisticated machine learning algorithms are expected. Kong et al. [42] developed ML-based models to predict the hospital mortality of sepsis patients based on the MIMIC-III database, utilizing demographic features and clinical features, such as acute physiological measurements in the first 24 h in ICU and chronic health status. They suggested that the gradient boosting machine model demonstrated the highest AUROC (0.845) and good calibration compared to other ML models and SAPS-II. In addition, a deep-learning-based autonomous pipeline was constructed, capable of offering hourly refreshed probabilities on sepsis patients’ survival [43]. This powerful tool incorporated a broad scope of structured or unstructured data in MIMIC-III, alleviating the workload from manual feature selection and data curation. The model showed good discrimination at 48 h (AUROC: 0.8463) and could guide ICU step-down strategies among low-risk patients. Recently, Hu et al. [14] developed an XGBoost model for early mortality prediction using MIMIC-IV, showing a higher AUROC of 0.884, compared to other ML models, such as Random Forest (0.882), Naive Bayes (0.856), etc. However, these models utilized a validation cohort for model training as well as performance assessment, which might cause data leakage and overestimate the actual performance. In this study, all XGBoost models performed significantly better than LR models using the same feature sets, and we further validated greatly improved discrimination in an independent test cohort compared with long-established scoring tools. This tool can help intensivists to identify high-risk patients early; thus, they can initiate therapies carrying more than minimal risk for younger patients [4], or consider symptom relief and end-of-life (EOL) issues for elderly patients [44]. The interpretation of the model should also be examined with caution, as the calibration of the model showed a tendency towards underestimation of risk, especially when observed risks rose. Therefore, patients who are classified as low risk by the ML model might still need EOL preparedness, when high-risk elderly patients and family members could benefit from understanding their own mortality and disease trajectories in early phases and advocating for care consistent with their goals and values [45].

Nevertheless, machine learning has not been so widely acknowledged and applied in the field of critical care as that in marketing, social media, and other domains [12]. Despite the emerging number of new publications, few examples of data-driven decision-supporting systems have been successfully run in hospital settings. One of the primary reasons is that models must be externally validated before clinical use. Data have shown that only 7% of newly published predictive tools conducted external validation in 2019 [17]. Temporal validation is considered to lie midway between internal and external validation. It would guarantee a reproducible satisfactory performance in new cohorts and various settings. Thus, with temporal validation and subgroup performance evaluation, our new model worked robustly. Another major issue is that clinicians could not trust the advice from such systems if they could not understand the rationale behind them. Hence, good interpretability is crucial not only for researchers to verify the reliability of new models, but also for clinicians to gain confidence in sharing decision-making responsibilities with algorithms [12]. From our interpretable ML model, we discovered a few compelling predictors significantly related to the survival of sepsis.

According to model interpretation, administering supplemental oxygen therapy can be a beneficial sign for sepsis patients, whereas implementing high-flow nasal cannula (HFNC) or non-invasive ventilation (NIV) was not significantly correlated with mortality according to ORs from the LR “Whole” model (*p* > 0.05). Oxygen supplementation has been commonly applied in patients with sepsis when insufficient trials are focused explicitly on different non-invasive oxygenation strategies in the sepsis cohort [46]. Urine output in the first day and minimum BUN both showed pivotal importance in predicting mortality, as sepsis-associated acute kidney injury (AKI) is strongly associated with poor clinical outcomes [47]. In sepsis, oliguria might carry increased significance in detection of AKI, possibly because it is tricky to confirm urine output criteria outside the ICU [48]. Urine output also demonstrated crucial importance in predicting short- and long-term survival, even a brief episode of isolated oliguria (no serum creatinine criteria present) seems to be associated with decreased 1-year survival [49]. In addition, maximum and minimum serum lactate were both presented as the top-20 critical factors contributing to sepsis prognosis based on feature importance. Serum lactate has been suggested as a decisive risk factor for sepsis patients, as Liu et al. [50] found that the discrimination of lactate was superior to quick-SOFA score and similar to SOFA score (AUROC: 0.664, 0.547, 0.686, respectively). Since Survival Sepsis Campaign’s 1 h bundle [51] has been broadly promoted, serum lactate has become a routine measurement in ICU; thus, introducing lactate levels into novel scoring systems should be advocated.

Compared to previous publications, our study showed several strengths. Firstly, we employed the up-to-date MIMIC-IV database to develop our models. The XGBoost model was significantly more efficient in learning high-order interactions and non-linear functions than conventional regression. Temporal validation was executed without data leakage, confirming the reliability of performance metrics. The model was also assessed among various subgroups, indicating a robust predictive ability in practical settings. In addition, good interpretability was intently inspected to provide clues for future research. 

Despite that, the current study also had the following limitations. It was a single-centered retrospective analysis, so potential bias was inevitable. In addition, clinical features were considered only within the first 24 h in the ICU, so progressive physiological status changes were not fully incorporated into the current model. Third, the model was not validated in external datasets to examine its generalizability in spatio-temporally diverse groups.

## 5. Conclusions

In conclusion, this study suggested an interpretable XGBoost-based model, and temporal validation further demonstrated that this tool performed significantly better than LR and outperformed conventional severity scores in predicting in-hospital mortality among sepsis patients and varying subgroups. Additionally, this model alleviated the “black-box” issue with good interpretability, indicating urine output, supplemental oxygen therapy, maximum norepinephrine rate, etc., as vital predictors of sepsis survival. 

## Figures and Tables

**Figure 1 jcm-12-00915-f001:**
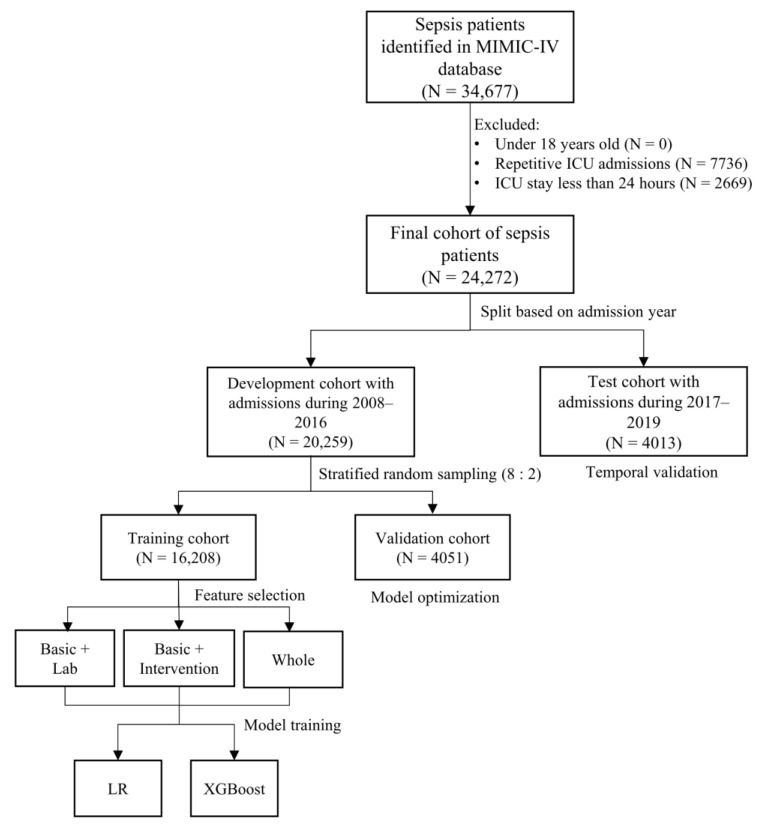
Flowchart of cohort selection.

**Figure 2 jcm-12-00915-f002:**
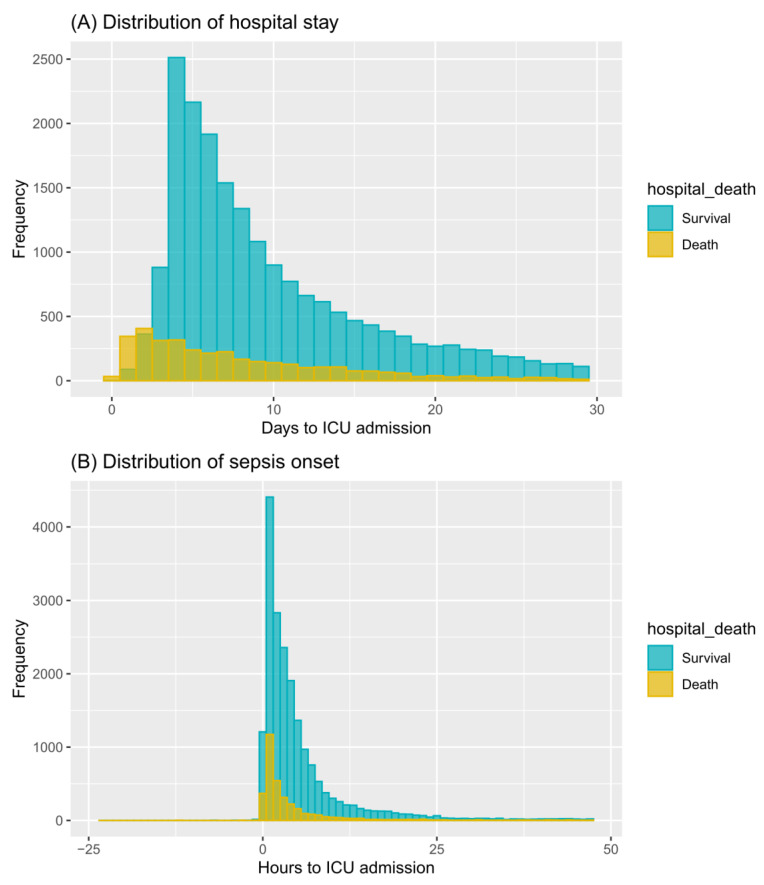
The distribution of hospital stays and sepsis onset after ICU admission according to hospital mortality in all patients (N = 24,272). Panel (**A**) illustrates the distribution of hospital length of stays over 30 days after ICU admission stratified by patients’ in-hospital outcomes; (**B**) shows the distribution of sepsis onset before 24 h and after 48 h of ICU admission among survivors and non-survivors. Time intervals were calculated using the timepoint of ICU admission as point zero. Blue bins represented survivors and yellow bins represented non-survivors.

**Figure 3 jcm-12-00915-f003:**
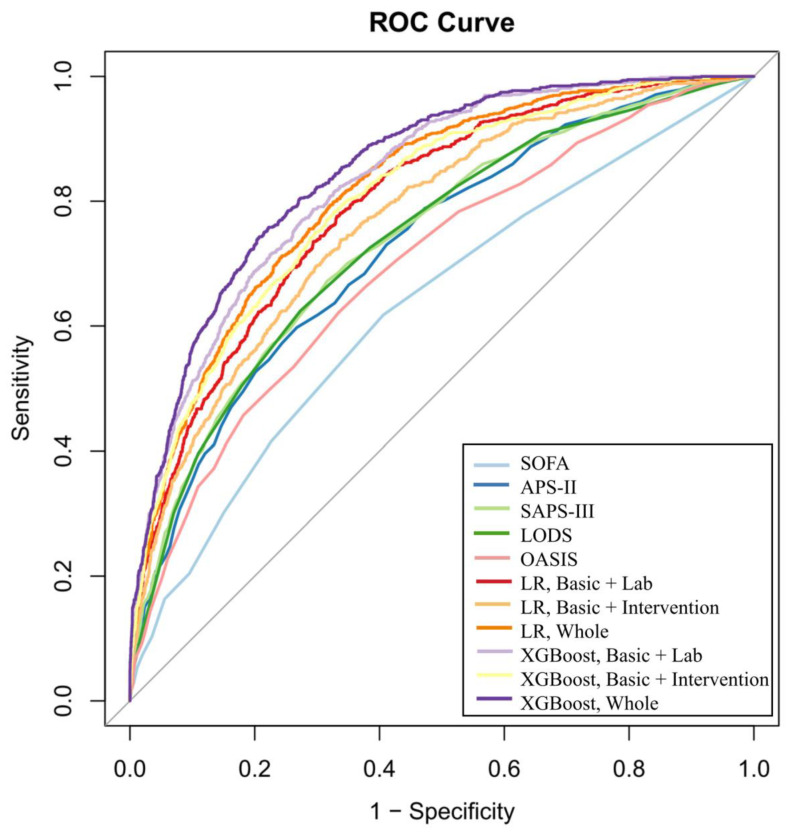
ROC curves presenting the performance of logistic regression, XGBoost models and single scores for predicting in-hospital mortality in the test cohort (N = 4013). ROC, receiver operating characteristics; SOFA, sequential organ failure assessment; SAPS, simplified acute physiology score; LODS, logistic organ dysfunction score; OASIS, oxford acute severity of illness score; XGBoost, extreme gradient boosting; LR, logistic regression.

**Figure 4 jcm-12-00915-f004:**
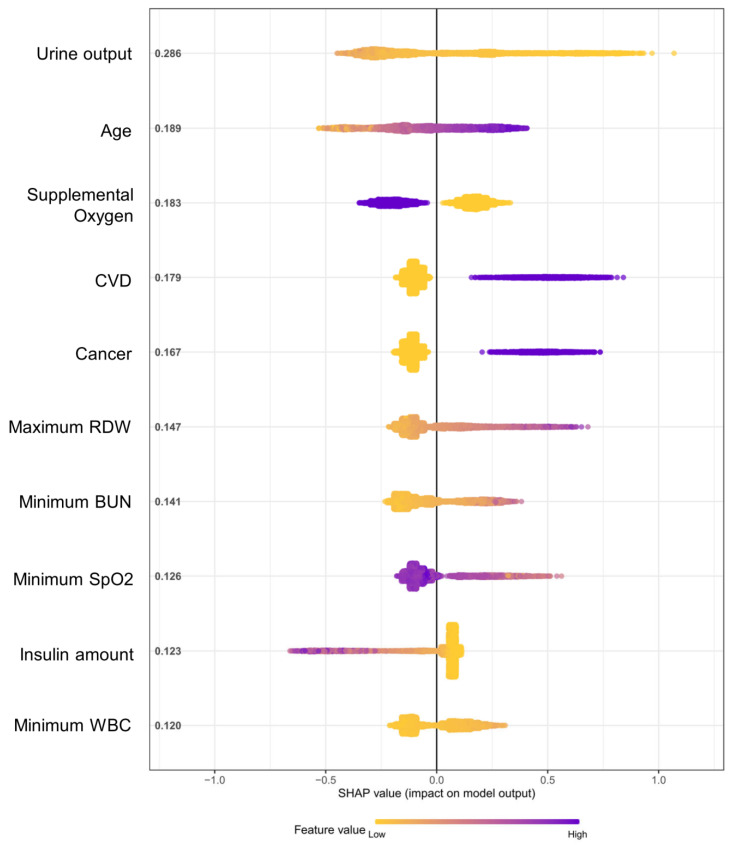
The SHAP summary plot of the XGBoost “Whole” model for predicting in-hospital mortality of sepsis in the test cohort (N = 4013) (top 10 features included). SHAP, Shapley Additive explanations; XGBoost, extreme gradient boosting; CVD, cerebrovascular disorder; RDW, red blood cell distribution width; BUN, blood urea nitrogen; SpO2, peripheral oxygen saturation; WBC, white blood cell.

**Figure 5 jcm-12-00915-f005:**
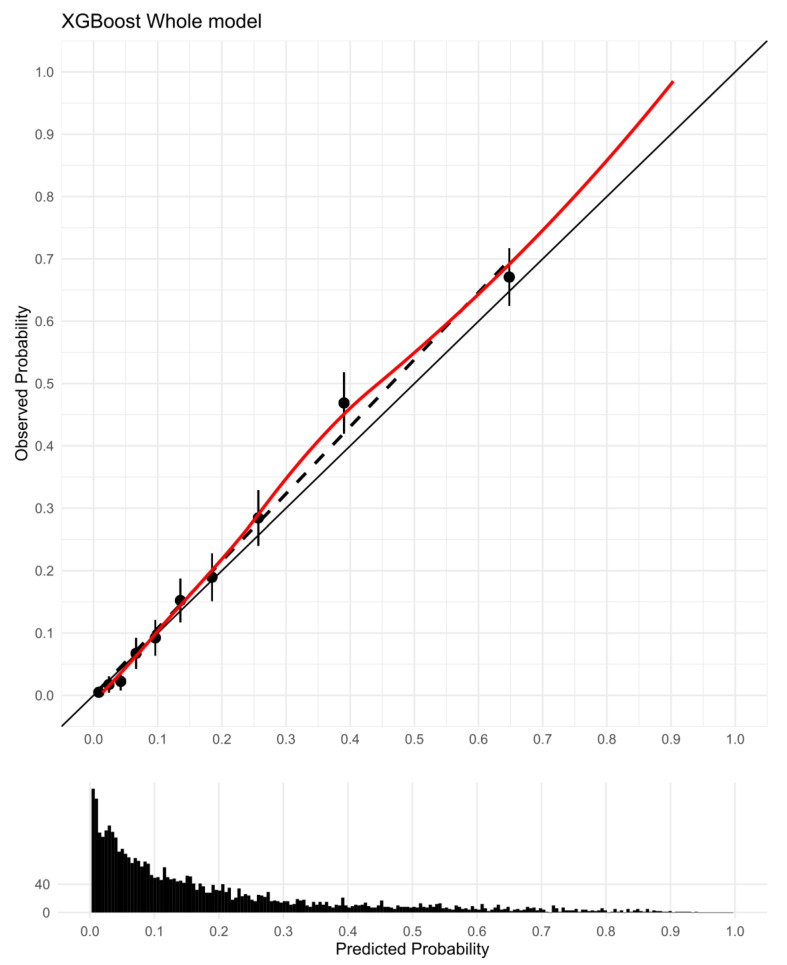
Calibration plot of the XGBoost “Whole” model in the test cohort (N = 4013). The black solid line at 45 degrees referred to a benchmark line of perfect calibration. The 10 black dots were average estimated and observed probabilities within 10% of the population divided based on predicted probability, and a black dashed line was straight through estimates. The red solid line represented a smoothed loess line through predicted outcomes.

**Table 1 jcm-12-00915-t001:** Baseline features of 24,272 sepsis patients categorized by in-hospital mortality.

Features	Total (N = 24,272)	Survivors (N = 20,513)	Non-Survivors (N = 3759)	*p*-Value
Male, %	14,060 (57.9%)	11,999 (58.5%)	2061 (54.8%)	<0.001
Age, Mean (SD)	66.1 (16.2)	65.4 (16.2)	69.7 (15.3)	<0.001
Race, %				
Other/Unknown	4225 (17.4%)	3268 (15.9%)	957 (25.5%)	<0.001
Asian	688 (2.8%)	585 (2.9%)	103 (2.7%)	0.74
African American	2074 (8.5%)	1773 (8.6%)	301 (8.0%)	0.21
Hispanic	831 (3.4%)	728 (3.5%)	103 (2.7%)	0.01
Caucasian	16,454 (67.8%)	14,159 (69.0%)	2295 (61.1%)	<0.001
BMI, Median (Q1, Q3)	27.7 (24.1, 32.4)	27.7 (24.2, 32.4)	27.0 (23.2, 32.3)	<0.001
Days in ICU, Median (Q1, Q3) *	3.2 (1.8, 6.5)	3.0 (1.8, 6.0)	4.7 (2.4, 9.2)	<0.001
Days in hospital, Median (Q1, Q3) *	7.9 (5.0, 14.2)	8.1 (5.2, 14.6)	6.5 (3.0, 12.9)	<0.001
Hours to initiate antibiotics, Median (Q1, Q3) *	4.2 (1.8, 16.4)	4.0 (1.8, 14.3)	5.1 (2.2, 28.0)	<0.001
Hours to take culture initiation, Median (Q1, Q3) *	−0.1 (−4.8, 3.1)	0.1 (−4.9, 3.2)	−0.9 (−4.4, 2.8)	0.953
Hours of sepsis onset, Median (Q1, Q3) *	3.1 (1.3, 6.8)	3.2 (1.4, 6.9)	2.0 (1.0, 5.8)	<0.001
Admitted ICU, %				
CCU/CVICU	7505 (30.9%)	6791 (33.1%)	714 (19.0%)	<0.001
MICU	5268 (21.7%)	4196 (20.5%)	1072 (28.5%)	<0.001
MICU/SICU	4473 (18.4%)	3575 (17.4%)	898 (23.9%)	<0.001
Neuro-ICU	894 (3.7%)	751 (3.7%)	143 (3.8%)	0.70
SICU	3361 (13.8%)	2835 (13.8%)	526 (14.0%)	0.80
TSICU	2771 (11.4%)	2365 (11.5%)	406 (10.8%)	0.21
Admission type, %				
Observation Admit	2284 (9.4%)	1864 (9.1%)	420 (11.2%)	<0.001
Emergency	13,128 (54.1%)	10,824 (52.8%)	2304 (61.3%)	<0.001
Elective	1152 (4.7%)	1108 (5.4%)	44 (1.2%)	<0.001
Surgical Admit	2599 (10.7%)	2523 (12.3%)	76 (2.0%)	<0.001
Urgent	5109 (21.0%)	4194 (20.4%)	915 (24.3%)	<0.001
Marital status, %				
Other	2173 (9.0%)	1542 (7.5%)	631 (16.8%)	<0.001
Divorced/Widowed	4807 (19.8%)	4020 (19.6%)	787 (20.9%)	0.06
Married	10,991 (45.3%)	9468 (46.2%)	1523 (40.5%)	<0.001
Single	6301 (26.0%)	5483 (26.7%)	818 (21.8%)	<0.001
Admitted period, %				
2008–2010	8390 (34.6%)	7205 (35.1%)	1185 (31.5%)	<0.001
2011–2013	6053 (24.9%)	5183 (25.3%)	870 (23.1%)	0.006
2014–2016	5816 (24.0%)	4902 (23.9%)	914 (24.3%)	0.60
2017–2019	4013 (16.5%)	3223 (15.7%)	790 (21.0%)	<0.001
Pre-existing disease, %				
MI	4332 (17.8%)	3533 (17.2%)	799 (21.3%)	<0.001
CHF	7303 (30.1%)	5940 (29.0%)	1363 (36.3%)	<0.001
CVD	3604 (14.8%)	2824 (13.8%)	780 (20.8%)	<0.001
CPD	6431 (26.5%)	5359 (26.1%)	1072 (28.5%)	0.002
CKD	5322 (21.9%)	4304 (21.0%)	1018 (27.1%)	<0.001
Diabetes	7432 (30.6%)	6299 (30.7%)	1133 (30.1%)	0.489
Cancer	3785 (15.6%)	2821 (13.8%)	964 (25.6%)	<0.001
Source of infection, %				
Bloodstream	2002 (8.2%)	1640 (8.0%)	362 (9.6%)	<0.001
Respiratory tract	2298 (9.5%)	1744 (8.5%)	554 (14.7%)	<0.001
Cerebrospinal fluid	25 (0.1%)	23 (0.1%)	2 (0.1%)	0.301
Gastrointestinal/Peritoneal	372 (1.5%)	291 (1.4%)	81 (2.2%)	<0.001
Genitourinary	1957 (8.1%)	1620 (7.9%)	337 (9.0%)	0.027
Other	1175 (4.8%)	1008 (4.9%)	167 (4.4%)	0.216
Disease severity score				
SOFA, Median (Q1, Q3)	3.0 (2.0, 4.0)	3.0 (2.0, 4.0)	4.0 (2.0, 5.0)	<0.001
LODS, Mean (SD)	5.2 (2.9)	4.8 (2.7)	7.4 (3.3)	<0.001
SAPS-II, Mean (SD)	39.5 (14.1)	37.5 (12.9)	50.3 (15.4)	<0.001
APS-III, Mean (SD)	49.8 (21.5)	46.6 (19.0)	67.2 (25.4)	<0.001
OASIS, Mean (SD)	33.2 (8.5)	32.2 (8.1)	38.7 (8.8)	<0.001

BMI, body mass index; CVICU, Cardiovascular Intensive Care Unit; CCU, Coronary Care Unit; MICU, Medical Intensive Care Unit; SICU, Surgical Intensive Care Unit; TSICU, Trauma Surgical Intensive Care Unit; MI, Myocardial infarction; CHF, Congestive Heart Failure; CVD, Cerebrovascular Disorder; CPD, Chronic Pulmonary Disease; CKD, Chronic Kidney Disease; SOFA, Sequential Organ Failure Assessment; LODS, Logistic Organ Dysfunction Score; SAPS, Simplified Acute Physiology Score; OASIS, Oxford Acute Severity of Illness Score. * Time intervals were calculated using the timepoint of ICU admission as point zero. Positive and negative values referred to events that happened after and before ICU admission, respectively.

**Table 2 jcm-12-00915-t002:** Performance metrics of logistic regression, XGBoost models and single scores in the training and test cohorts.

Models	AUROC	AUROC 95% CI	Cut-Offs	Se	Sp	Accuracy	PPV	NPV	DOR
Training cohort (N = 16,208)									
XGBoost, Basic + Lab	0.97	[0.964, 0.971]	0.23	0.89	0.92	0.92	0.66	0.98	95.12
XGBoost, Basic + Intervention	0.90	[0.892, 0.905]	0.15	0.86	0.78	0.79	0.40	0.97	21.56
XGBoost, Whole	0.95	[0.942, 0.950]	0.18	0.90	0.85	0.86	0.51	0.98	51.00
LR, Basic + Lab	0.82	[0.815, 0.832]	0.13	0.77	0.72	0.73	0.32	0.95	8.94
LR, Basic + Intervention	0.81	[0.796, 0.814]	0.14	0.75	0.72	0.72	0.31	0.94	7.04
LR, Whole	0.84	[0.836, 0.852]	0.13	0.81	0.72	0.73	0.33	0.96	11.82
Test cohort (N = 4013)									
XGBoost, Basic + Lab	0.83	[0.817, 0.847]	0.16	0.77	0.73	0.73	0.41	0.93	9.23
XGBoost, Basic + Intervention	0.80	[0.788, 0.821]	0.15	0.77	0.69	0.70	0.37	0.92	6.75
XGBoost, Whole	0.85	[0.835, 0.863]	0.20	0.76	0.78	0.77	0.45	0.93	10.87
LR, Basic + Lab	0.80	[0.779, 0.813]	0.18	0.78	0.67	0.69	0.36	0.92	6.47
LR, Basic + Intervention	0.77	[0.751, 0.787]	0.18	0.74	0.66	0.68	0.35	0.91	5.44
LR, Whole	0.82	[0.800, 0.832]	0.18	0.80	0.67	0.70	0.38	0.93	8.14
SOFA	0.63	[0.609, 0.653]	3.5	0.62	0.59	0.60	0.27	0.86	2.27
SAPS-II	0.73	[0.708, 0.747]	44.5	0.60	0.73	0.71	0.35	0.88	3.95
APS-III	0.74	[0.719, 0.758]	52.5	0.67	0.68	0.68	0.34	0.89	4.17
LODS	0.74	[0.716, 0.755]	6.5	0.62	0.73	0.71	0.36	0.89	4.55
OASIS	0.70	[0.676, 0.717]	35.5	0.62	0.67	0.66	0.31	0.88	3.29

AUROC, area under the receiver operating characteristic curve; CI, confidence interval; Se, sensitivity; Sp, specificity; PPV, positive predictive value; NPV, negative predictive value; DOR, diagnostic odds ratio; XGBoost, extreme gradient boosting; LR, logistic regression; SOFA, sequential organ failure assessment; SAPS, simplified acute physiology score; LODS, logistic organ dysfunction score; OASIS, oxford acute severity of illness score.

**Table 3 jcm-12-00915-t003:** Subgroup performance of the “Whole” XGBoost model for predicting in-hospital mortality in the test cohort (N = 4013).

Subgroup	No. of Patients	Prevalence of Non-Survivors	AUROC	AUROC 95% CI
Admission type				
Observation Admit	896	18.6%	0.82	[0.792, 0.858]
Emergency	1712	21.6%	0.84	[0.820, 0.861]
Elective	163	1.8%	0.98	[0.930, 1.000]
Surgical Admit	193	8.8%	0.83	[0.715, 0.953]
Urgent	1049	22.2%	0.85	[0.824, 0.878]
ICU Unit Type				
CCU/CVICU	1008	14.9%	0.88	[0.855, 0.908]
MICU	884	25.6%	0.86	[0.832, 0.886]
MICU/SICU	653	22.7%	0.79	[0.750, 0.825]
Neuro-ICU	582	16.8%	0.81	[0.769, 0.859]
SICU	399	18.0%	0.87	[0.826, 0.909]
TSICU	487	19.7%	0.85	[0.803, 0.888]
Source of infection				
Bloodstream	353	26.1%	0.76	[0.703, 0.819]
Respiratory tract	426	29.3%	0.72	[0.668, 0.771]
Gastrointestinal/Peritoneal	74	28.4%	0.72	[0.605, 0.843]
Genitourinary	421	22.1%	0.79	[0.738, 0.838]
Other	177	13.6%	0.79	[0.693, 0.888]

XGBoost, extreme gradient boosting; area under the receiver operating characteristic curve, AUROC; CI, confidence interval; CVICU, Cardiovascular Intensive Care Unit; CCU, Coronary Care Unit; MICU, Medical Intensive Care Unit; SICU, Surgical Intensive Care Unit; TSICU, Trauma Surgical Intensive Care Unit.

## Data Availability

Restrictions apply to the availability of these data. Data was obtained from PhysioNet and are available at https://physionet.org/content/mimiciv/2.1/ (accessed on 27 December 2022) with the permission of PhysioNet.

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
