# Peer review of "Developing an Interpretable Machine Learning Model to Predict in-Hospital Mortality in Sepsis Patients: A Retrospective Temporal Validation Study"

_jcm, 2023, doi:10.3390/jcm12030915_

Round 1

Reviewer 1 Report

The aim of the paper by Li et al. was to develop an interpretable machine-learning model to predict in-hospital mortality in patients with sepsis. The performances of the models developed using XGBoost and logistic regression were compared with several established severity scores. The authors conclude that XGBoost showed the highest discriminatory power compared to logistic regression and severity scores. Although interesting results, the current version of the manuscript lacks detailed information on both methods and results. This makes it difficult to fully assess the scientific soundness of the study. The novelty of the work is also poorly described as others have developed similar prediction models with a performance equal to the XGBoost model. From my point of view, the manuscript is presented in a well-structured manner and its major strength is the large sample size used for training and test of the models.

General comments

1.       Too many decimal places used e.g., for OR, AUC-values, and 95% CI. Review and correct throughout the manuscript.

2.       In accordance with scientific writing, expressions such as “Figure 2 shows” or “As can be seen in Table 3” should be avoided. References in the text to figures and tables should be within brackets as in “The highest rainfall is usually received in April (Table 1).” In the manuscript, for e.g., on line 173 “ Figure 1 presented a total of…”, should instead be written as “ A total of 24,573 patients fulfilling the inclusion criteria in MIMIC-I, of whom 15.4% died in hospital (n=3,785) (Figure 1).”

3.       Sometimes the term “variables” are used, and other times, the term “features” are used. Better to be consistent and only use one term to denote the same thing.

4.       Too many abbreviations are used throughout the manuscript. Keep abbreviations to a minimum. Do not abbreviate a term or phrase that is used only once or twice in the paper. Only use abbreviations for terms or phrases that are frequently used.

Major comments

1.      The problem is not clearly explained in the introduction - what is the motivation for predicting mortality? How can the result be useful for the hospitals?

2.      There are more references to be included in the introduction regarding machine learning for sepsis diagnosis and prognosis, and also regarding the interpretability of machine learning for clinical use.

3.      Ethics statement is missing.

4.      A description of the types of missing data (MCAR, MAR or MNAR) is missing.

5.      Regarding the handling of missing data, only features with more than 50% of missing values were removed. Although there is no established cut-off regarding an acceptable % of missing data, 50% appear too high, especially when working with biological data with implications for clinical decision. Besides, any references supporting the choice of 50% and how it affects the outcome are not provided.

6.      The outcome variable, i.e., in-hospital mortality, is not clearly described. Did you include all deaths occurring in hospital, no matter how many days since admission? What was the median (IQR) hospital stay for the patients who died, i. e., on what day did they die? Are there any differences in measured features depending on which day the death occurred?

7.      Why not split the cohort into training, validation and test datasets?

8.      Were the two groups (survivors and non-survivors) sampled separately to preserve the ratio in training and test?

9.      No reflections are made on how an imbalanced test data set affects the evaluation parameters. It is well-known that sensitivity, specificity and accuracy are related to prevalence in a population. It would be good to also use some evaluation parameters that are less dependent on the prevalence, such as the diagnostic odds ratio.

10.   Line 215: here you write “performance was unsatisfactory in the test cohort due to incapacity of processing missing values”. I thought that you have imputed data also for the test data set? Usually, one handles the missing data before the split into training and test so this confuses me. 

11.   A description of the different severity scores employed is missing, e.g., what variables are included for which score and which cut-off values are used? Not all scores are used in all countries of the world, there are many regional variations, so it is important to describe this clearly. Please include this in the manuscript as well.

12.   No information is given about which or how many features were selected for each model nor which features and corresponding cut-off values were included for each severity score. Please include this in the manuscript as well.

13.   No performance results for the severity scores are shown. Please include this in the manuscript as well.

14.   No statistical comparison is made between AUC, I recommend the authors to use DeLong’s test for that purpose. Then you can say that one AUC is significantly better than another. If multiple comparisons are made, then the obtained p-values should be adjusted for that as well.

15.   Not "fair" to compare XGBoost model with logistic regression model or with severity scores on a dataset containing missing data. It would be better to compare it against other models capable of handling missing data. Another solution would be to only test the LR model and the severity scores on patients with complete data. 

Minor comments

1.      Abstract (lines 46-47): it is not clear for which model/score each AUC represents.

2.       Line 83: use the abbreviation for machine learning.

3.       Line 92-93: information about the country is missing.

4.       Line 104: 2.2 Features and outcome – difficult to read this section. No complete sentences.

5.       Line 118: here you say that the outcome was the probability of in-hospital death of every sepsis patient. More correct is to say that it was the model outcome, the variable you aimed to predict. The dependent outcome variable in your dataset is whether the patient died during his/her hospital stay.

6.       Line 174: do not start a sentence with a numeral. One should use spelled-out numbers instead at the beginning of a sentence in place of numerals. This distinction is not based on grammar, but rather on the conventions of academic writing in English.

7.       Line 178: what do you mean by potential bloodborne infection, what is the definition and how was potential bloodborne infection detected? Quite confusing. Besides, sepsis is not equivalent to bloodborne infection (=disease that can be spread through contamination by blood and other body fluids).

8.       Line 180: lots of numbers here but not clear what they represent. Mean, median, max, min? This goes for many more lines in the manuscript, e.g., lines 183, 184, 187, etc. Please look over the whole manuscript.

9.       Table 1: confusing. A mix of different types of numerals is used – it is not clear whether they represent total numbers, %, median/mean, IQR, max, min, etc. This table needs to be made clearer. In addition, thousands should always be separated by a comma, i.e., 14,222 instead of 14222 according to conventional English writing. The font size is too small in the table, challenging to read.

10.   Figure 1: it appears strange that you first have 17,177 patients in the training cohort, and that after oversampling the sample size has decreased to 10,572 patients. One expects the sample size to increase if one synthesizes data. Please explain. Another thing is that the textboxes start with numerals. Please reformulate e.g., Sepsis patients identified in MIMIC-IV database (n=35,010).

11.   Figure 2: the figure labels are too small, not readable.

12.   Figure 3: include information about the sample size for the test cohort, i.e., no. of patients in each group. Which dataset was used to generate the figure? Training or test?

13.   Figure 5: explain the figure more in the figure legend. Which dataset was used to generate the figure? Training or test? Sample size?

Author Response

Dear Editor and Reviewers,

Thanks for your efforts in reviewing the article and providing such insightful suggestions. I have revised the manuscript according to these comments. I will elaborate on the changes point by point in the letter below.

Reply to reviewer 1:

  1. All the issues about the statistical exhibition (e.g. too many decimals, the use of “Figure and table”) and manuscript writing (e.g. the use of abbreviations) in “general comments” have been corrected.
  2. The introduction section has been rewritten to emphasize the meaning of mortality estimation and machine learning interpretability.
  3. The ethics statement has been added in the method section “2.1 Database and ethical approval”.
  4. The majority of the missing data are not frequently-ordered laboratory measurements. Features were only included within the first day in ICU, so doctors might think that early prescribing of these tests was not necessary considering the patients’ pathophysiological conditions. These features can be classified as missing at random (MAR). The previous setting of 50% missing as a threshold is a little high, and I reset it to 40% because Jakobsen et al. mentioned that above 40% missingness might lead to “hypothesis generating results” (10.1186/s12874-017-0442-1). However, liver function tests such as alanine transaminase and total bilirubin were preserved despite the missing proportions of 41-42%, because dropping these features would cause a huge neglection of liver function assessment.
  5. In-hospital mortality was defined as the all-cause mortality occurring in the hospital after the first day in the ICU. The distribution plot of the cohort's hospital stays and sepsis onset stratified by mortality has been supplemented (Figure 2). The majority of patients had hospital stays of less than 30 days (94%), when 78% of non-survivors died within 14 days.
  6. We changed the cohort selection process, and a holdout test dataset for temporal validation has been created using patients being treated from 2017 to 2019 in the first place.
  7. Stratified random sampling has been performed based on hospital mortality to generate training and validation cohorts for model development.
  8. We evaluated models using more metrics, including AUROC, sensitivity, specificity, accuracy, positive predictive value, negative predictive value, and diagnostic odds ratio.
  9. Originally, we split the dataset before imputation in order to preserve the test dataset comparable to real-world settings. We changed the cohort selection process and used an independent test dataset without data leakage for temporal validation now.
  10. Cut-off values and performance indexes of severity scores have been updated in the revised version (Table 2).
  11. Statistical comparison between AUROC has been updated in the section “3.2 Model performance”.
  12. We changed the cohort selection process and trained XGBoost and logistic regression models in the complete dataset of the development cohort, and further validated in an independent test dataset.
  13. The writing issues mentioned in the “minor comments” have all been corrected.
  14. The bloodstream infection was defined as positive blood culture results. We added subgroups with varying sources of infection to test our models.

Reviewer 2 Report

This study investigates in-hospital mortality prediction of ICU patients diagnosed with sepsis using a machine-learning model based on a single-center cohort of patients. While it is an interesting study based on machine-learning techniques, well written, and easy to follow, the lack of external validation is the main limitation of the work.

- Please mention that while the observation window is the first 24 hours of ICU admission, what is the prediction window? All the patients died after 24 hours of admission? Have you excluded dead patients in the first 24 hours?

- Could you please provide the distribution plot of the cohort's mortality days and sepsis onset time? All the sepsis onset times happened within the first 24 hours of ICU admission? 

- Did you use stratified random sampling with keeping positive prevalence in the test set?

- Difference in the AUC of the model in training and test set could be due to model overfitting?

- What was the model calibration performance? Please provide the reliability diagram for the methods.

- Please compare your study with this article DOI: 10.1007/s40121-022-00628-6, mentioning similarities and differences.

Author Response

Dear Editor and Reviewers,

Thanks for your efforts in reviewing the article and providing such insightful suggestions. I have revised the manuscript according to these comments. I will elaborate on the changes point by point in the letter below.

Reply to review 2:

  1. Although external validation was not performed in this study, we utilized temporal validation instead, i.e., only including patients admitted from 2017 to 2019 in an independent test dataset, when trained out models using patients from earlier periods.
  2. The issue of the prediction window is quite crucial. When extracting eligible patients from the database, we added a filtering condition that only patients with ICU length of stay >= 1 day were included. Therefore, patients who died within 24 hours were not considered in the beginning. I have highlighted the prediction window in the method section.
  3. The distribution plot of the cohort's hospital stay and sepsis onset stratified by mortality has been supplemented (Figure 2). 91.5% (22,202) of all patients had sepsis attacks in the first 24 hours, and non-survivors experienced a more acute sepsis onset than survivors following ICU admission (2.0 vs. 3.2 hours, P < 0.001)
  4. Stratified random sampling has been performed based on hospital mortality in order to generate training and validation cohorts.
  5. We used different feature selection strategies and hyperparameter tuning in order to avoid overfitting. XGBoost and LR models mainly performed well in both development cohort and temporal validation.
  6. The calibration plot for our XGBoost model has been supplemented (Figure 5).
  7. The comparison of this publication has been appended in the discussion section. In sum, despite the similarities in database and model selection, this study used an up-to-date version of the database, included more details regarding clinical interventions, utilized temporal validation, and verified the robustness of model performance across varying subgroups compared to the article.

Round 2

Reviewer 1 Report

It was quite tricky to review this revision as no point-by-point response to my comments was provided. In conclusion, I believe that the scientific work has been greatly improved, but there are still some minor shortcomings (see my comments below). In addition, the authors seem to overestimate the results obtained. I would rather say that the performance of their model is not superior to logistic regression model, but rather slightly better. Still, some of my major comments have not been addressed fully:

·       A description of the different severity scores employed is missing, e.g., what variables are included for which score and which cut-off values are used? Not all scores are used in all countries of the world, there are many regional variations, so it is important to describe this clearly. Please include this in the manuscript as well.

·       I think it is not fair to compare XGBoost model with logistic regression model nor with severity scores on a dataset containing missing data. It would be better to compare it against other models capable of handling missing data. Another solution would be to only test the LR model and the severity scores on patients with complete data.

I hope the authors do not feel overwhelmed by my comments below. They are minor, my only intention is to help the authors to further improve the manuscript.

Abstract

Line 35: …management, when existing … … management, as existing …

Line 36: … and validated a … validate a …

Line 36: … to estimate in-hospital … to predict in-hospital …

Line 44: …24,272 patients, and 4,013 patients formed the test cohort for temporal validation. Not clear whether the 4,013 patients were included in 24,272 patients or not. Better to formulate it like “…24,272 patients whereof 4,013 patients formed the test cohort for temporal validation.”

Line 52: Superior performance – not correct to say that XGBoost was superior as its AUC was 0.85 and LR “Whole model” 0.82. I would say it performed better, but not superior. This goes for the whole manuscript.

Introduction

Line 148: … to estimate in-hospital mortality … … to predict in-hospital mortality

Line 187: …Patients under 18 years old, having repetitive ICU … Patients under 18 years old, or having repetitive ICU

Methods

Line 251-252: The R packages are not correctly cited. Moreover, it is important to acknowledge all the R-packages used and their developers., not just writing etc.

Line 254-255: The authors mention that they have checked the normality distribution and, based on that, described the variables using mean (SD) or median (IQR). So far so good, but then they say that they have used either Student´s test (parametric test for comparing two independent groups having a normal distribution) or Kruskal Walli´s test (non-parametric test for comparing more than two groups). Makes no sense?

Line 260: How many features had >40% missing data and were subsequently removed from further analysis? It is mentioned 20 features in line 415, but is this figure with or without the features associated with liver function?

Results/Discussion

Line 634: …that estimating in-hospital… … that predicting in-hospital…

Lines 738, 771: I do not agree about “superior discrimination” and “outperformed”, see my previous comment above.

Lines 746-747, 768, 860: previously you have used the term AUROC, not AUC, and with fewer decimal places. Correct this for consistency.

Table 1: In the Method section, you say that you describe some data using median and IQR. However, in this table, such data is described using the median and Q1 and Q3. Please correct to match.

Figure 2: Separate descriptions of parts A and B in the legend are missing. In addition, add information about the exact number of patients included instead of writing “all patients”.

Table 2/Figure 1: In Table 2, results of the performance of the whole development cohort are presented, whereas, in the flowchart in figure 2, it can be seen that the developmental cohort was divided into a training dataset and a validation dataset. Usually one uses the training cohort for the development of the models, the validation cohort to evaluate the performances of the developed models and then select the best model based on the results from the validation cohort. Finally, the selected model is evaluated on a test cohort to confirm its performance. It is not clear to me how and why another approach was applied here. 

Figure 5: lacks a description of what the different lines represent.

A general comment on the figure legends is that they should be more descriptive. One should be able to understand a figure fully just by looking at it and reading the legend. It should not require reading the main text. /L   

Table S1: I think that the Unit for some features suffers from typos: e.g., White blood cell 109/L should be 109/L.

Author Response

Thanks for your efforts in reviewing the article and for providing such insightful suggestions to improve this work. Really sorry for the trouble caused by summarizing replies towards several issues into one instead of offering replies point by point last time.

1st revision comments

  1. The description of various severity scores and cut-off values used in the manuscript has been supplemented in Table S2.
  2. In the original version, we split the dataset before imputation in order to preserve the test dataset comparable to real-world settings. After the 1st revision, we changed the cohort selection process and imputed the whole dataset followed by splitting it into the train-validation-test cohort. Therefore, we made a comparison between multiple models on the test cohort with complete data (N=4,013).

Abstract

The writing issues that the reviewer mentioned in lines 35, 36, and 44 have been corrected.

Line 52: After the DeLong test, we found that the AUROCs of XGBoost models are significantly higher than all LR models, although the differences are not very large. We change the word “superior” to “significantly better/improved” in the manuscript in order to keep the expression precise.

Introduction

The writing issues in lines 148 and 187 have been corrected.

Methods

Line 251-252: We included all wielded R packages and citations in the section “2.3 Statistical analysis and missing data processing” (Line 128-130).

Line 254-255: The previous mention of Kruskal Walli´s test was incorrect, it should be Wilcoxon rank-sum test if the continuous variable does not meet normal distribution. In fact, the function “tableby” in R package “arsenal” used in this work can set numeric tests as either ANOVA or Kruskal-Wallis test, which are equivalent to the t-test with equal variance or Wilcoxon rank-sum test when comparing two samples.

Line 260: We updated the descriptions in the section “3.1 Patient characteristics and missingness” (Line 187-189). We did not include features of maximal vasopressor rate last time, as we thought that those missing values actually didn’t exist due to non-usage. In fact, a total of 26 features had missing proportions above 40%, of which 8 features relating to liver function and 6 features of maximal vasopressor rate were preserved and imputed, and 12 features were subsequently discarded.

Results/Discussion

The writing issue in line 634 has been corrected.

Line 738, 771: We change the word “outperform” to “the performance was significantly better or lower” when comparing XGBoost and LR models. The word “outperform” was preserved when comparing XGBoost and severity scores because of the large gaps between AUROCs.

Lines 746-747, 768, 860: We keep the term “AUROC” consistent in the article.

Table 1: We updated the description in the section “2.3 Statistical analysis and missing data processing” (Line 130).

Figure 2: The description of panels A and B and patient numbers in Figure 2 have been supplemented in the legend.

Table 2/Figure 1: The process described here matches the actual process of ML model development in this research. We initially thought the data of the validation cohort was also utilized in model development for hyperparameters optimization and performance evaluation, thus we mentioned the development cohort instead of the training cohort in Table 2. We have changed it to the training cohort in Table 2.

Figure 5: A description of the calibration plot has been added to the legend and we further clarified the results of Figure 5 in section “3.3 Interpretability and calibration” (Line 225-228).

Table S1: The unit issues have been corrected.

Many thanks for your time!

Author Response

No more comments from the second reviewer.
